# SeqAffordSplat: Scene-level Sequential Affordance Reasoning on 3D Gaussian Splatting

## Abstract

3D affordance reasoning, the task of associating language instructions with the functional regions of 3D objects, is a critical capability for embodied agents. With its photorealistic rendering and precise geometric fidelity, the recently emerged 3D Gaussian Splatting (3DGS) has become an ideal representation for such fine-grained localization. Despite its potential, existing 3DGS-based methods are confined to single-object and single-step interactions, failing to address the long-horizon, multi-object tasks common in the real world. To fill this gap, we introduce a novel task of Sequential 3D Gaussian Affordance Reasoning and construct SeqAffordSplat, the first large-scale dataset with over 1,800 complex scenes to support this research. We then propose SeqSplatNet, an innovative end-to-end framework that leverages a Large Language Model (LLM) for autoregressive planning, directly mapping high-level instructions to a sequence of precise 3D affordance masks. To enhance performance, we introduce a Conditional Geometric Reconstruction pre-training strategy to build a robust geometric prior and a Semantic Feature Injection mechanism to fuse multi-scale semantic knowledge from 2D Vision Foundation Models. Extensive experiments demonstrate that our model achieves state-of-the-art performance on our new benchmark, successfully advancing affordance reasoning for long-horizon and scene-level sequential tasks.

## 1 Introduction

3D affordance reasoning, which identifies functional regions on objects to enable specific actions in 3D space, is a fundamental perceptual capability for embodied agents (Deng et al., 2021; Yang et al., 2023). By linking perception and action, it underpins essential functionalities in a spectrum of applications, including robotic manipulation (Yamanobe et al., 2017), augmented reality (Steffen et al., 2019; Nagarajan & Grauman, 2020), and virtual reality (Dalgarno & Lee, 2010; Venkatakrishnan et al., 2023). This has motivated early explorations into affordance reasoning using point cloud representations (Yang et al., 2023; Li et al., 2024b; Deng et al., 2021). While these approaches demonstrate potential in predicting affordances from 3D geometry, they are often constrained by the inherent sparsity and discrete nature of point clouds, which impedes their ability to capture the fine-grained, continuous structures essential for precise interaction.

Driven by the high-fidelity representations of 3D Gaussian Splatting (3DGS) (Kerbl et al., 2023), the transition from sparse point clouds to 3DGS has drawn increasing attention for 3D scene understanding (Mohammadi et al., 2023; Zhu et al., 2024). The pioneering work (Wei et al., 2025) demonstrates promising improvement for precise affordance reasoning in 3DGS scenes, suggesting its superiority in capturing fine-grained affordance detail. However, this method targets a specific and controlled task prototype, where each scene consists of a single instance and each instruction requires just a single atomic action for execution.

Due to the inherent high-level succinctness of real-world instructions, 3D affordance reasoning requires both (i) the composition of ordered primitive instructions (each corresponding to a primitive action), and (ii) the dynamic shift of functional regions across object instances in complex scenes. As illustrated in Fig. 1, an instruction such as *operating a microwave to warm up food in a bowl* necessitates multiple interdependent actions across three distinct functional regions from different instances—a composite capability beyond current methods due to their constrained task prototypes. This reveals a critical gap in 3D affordance reasoning: the absence of a task formulation for sequen-

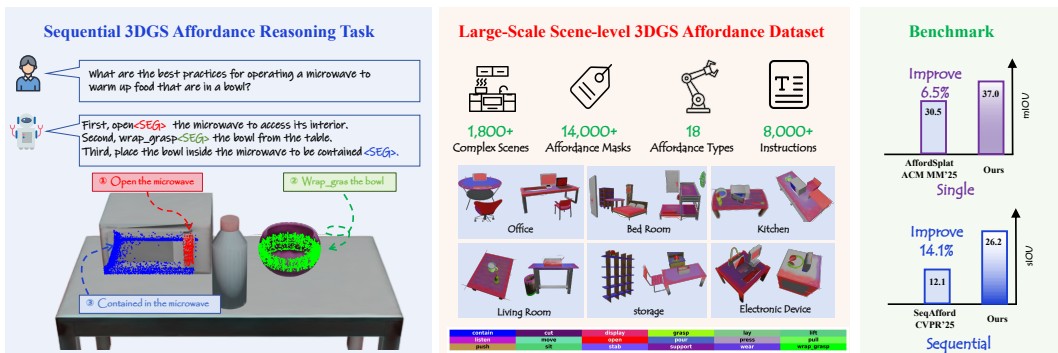

Figure 1: (**Left**) We introduce **Sequential 3DGS Affordance Reasoning Task** for complex, multi-step agent interactions. (**Center**) To support this, we present **SeqAffordSplat**, a large-scale scene-level dataset with over 1,800 3DGS scenes and 8,000 sequential instructions. (**Right**) Our model, **SeqSplatNet**, sets a new state-of-the-art, improving performance by 6.5% on single-step tasks and 14.1% on our sequential benchmark. Please zoom in for better visual effects.

tial interaction in cluttered scenes. To bridge this gap, we introduce the **Sequential 3D Gaussian Affordance Reasoning** task—a novel prototype designed for long-horizon and succinct instructions in complex environments with multiple functional regions and distractor objects, fundamentally advancing beyond prior object-centric, single-action approaches.

To facilitate research in this new direction, we introduce **SeqAffordSplat**, the first comprehensive benchmark designed for long-horizon, scene-wide affordance reasoning on 3DGS. The benchmark features a new large-scale dataset containing over 1,800 complex scenes, 14,000 affordance masks, and 8,000 sequential instructions.

To achieve sequential affordance reasoning, we introduce **SeqSplatNet**, the first framework designed for this novel and challenging task. Our approach uniquely integrates the hierarchical planning capabilities of Large Language Models (LLMs) with the rich representational power of 3DGS in a unified end-to-end architecture. Given a succinct instruction specifying a long-horizon action sequence, our model reasons about the user intent and grounds a sequence of actionable affordance masks directly onto the 3DGS scene representation. To facilitate learning 3DGS encoder from limited annotations, we design a Conditional Geometric Reconstruction Pre-training strategy, where the model reconstructs complete affordance regions guided only by an abstract semantic embedding of that region, thereby establishing a robust geometric prior. Furthermore, to resolve semantic ambiguities inherent in geometric structures, we design a Semantic Feature Injection mechanism that lifts rich semantic features from a frozen 2D Vision Foundation Model (VFM) via multi-view rendering and fuses them with the encoded geometric features at multiple scales during affordance mask decoding. This unified framework bridges the critical gap between high-level task planning and fine-grained 3D perception in complex environments. Our main contributions are summarized as follows:

- We introduce the new task of Sequential 3D Gaussian Affordance Reasoning and develop SeqAffordSplat, the first large-scale benchmark with over 1,800 3DGS scenes and 8,000 sequential instructions.

- We propose SeqSplatNet, a framework to unify high-fidelity 3DGS representation, long-horizon sequential planning, and complex scene-level understanding.

- We demonstrate through extensive experiments that our approach achieves a 14.1% performance improvement over sequential baselines on our challenging new benchmark.

## 2 RELATED WORK

### 2.1 AFFORDANCE LEARNING

Affordance Learning focuses on identifying functional regions in a scene, driven by either a closed-set of action types or open-vocabulary language instructions. This originates from 2D image

affordance segmentation, where each pixel is assigned to a predefined affordance category (Do et al., 2018; Roy & Todorovic, 2016). To generalize to unseen affordance types, recent approaches integrate Vision-Language Models (VLMs), aligning language instructions with visual affordances (Chen et al., 2025; Li et al., 2024a; Qian et al., 2024). However, these image-based approaches intrinsically lack the ability to capture explicit 3D spatial information—a critical requirement for robotic manipulation applications.

To overcome the limitations of image-based perception, researchers turn to 3D representations for accurate geometry awareness in 3D space. Pioneering studies (Deng et al., 2021; Xu et al., 2022; Mo et al., 2022) established benchmarks for affordance segmentation on 3D point clouds for predefined affordance type sets. This foundation later has evolved toward open-vocabulary affordance reasoning, where models identify affordance regions in response to language instructions by leveraging the cross-modality reasoning capability of foundational models (Li et al., 2024b; Xu et al., 2022; Shao et al., 2025; Lu et al., 2025). Despite this evolution, these approaches still struggle with long-horizon reasoning, which stems from their constrained task prototype that each instruction involves a single action.

To address this limitation, we highlight sequential affordance reasoning as a more practical task prototype, where each instruction maps to a sequence of atomic affordances. Compared with the contemporary SeqAfford (Yu et al., 2025), which is capable of generating sequential affordance masks on 3D point clouds, our SeqSplatNet overcomes its inefficiency in localizing fine-grained affordance regions through high-fidelity 3DGS representation.

## 2.2 Affordance Learning on 3DGS

Beyond 3D point clouds, 3DGS offers a more expressive representation through its explicit point-based structure and photorealistic, real-time rendering capabilities (Kerbl et al., 2023). These characteristics facilitate the development of various embodied AI systems (Zheng et al., 2024; Shorinwa et al., 2024; Lu et al., 2024), as 3DGS enables both instantaneous environmental perception and direct association of semantic information with spatially precise geometric locations.

In affordance learning, 3DAffordSplat (Wei et al., 2025) established the first large-scale benchmark for affordance reasoning using 3DGS. While notable for its substantial instance volume and diverse affordance coverage, this benchmark adopts a constrained task prototype where each instruction maps exclusively to a single discrete affordance mask, inherently omitting sequential reasoning pathways essential for complex manipulation scenarios.

In summary, existing approaches remain confined to either sequential reasoning on sparse point clouds with inadequate localization fidelity or single-step interactions on 3DGS without action sequencing capability. To the best of our knowledge, we pioneer the first framework for scene-level affordance learning in 3DGS that simultaneously achieves fine-grained affordance localization and sequential reasoning.

## 3 Task Definition and Dataset

### 3.1 Task Definition

Sequential 3D Gaussian Affordance Reasoning is the challenging task of interpreting a succinct input instruction, decomposing it into an ordered sequence of primitive actions, and identifying the corresponding step-wise affordance regions within a complex 3DGS scene.

Specifically, consider a 3D scene represented by a 3DGS model $\mathcal{G} = \{G_i\}_{i=1}^N$ comprising $N$ Gaussian primitives. Each primitive $G_i$ is parameterized by its position, opacity, scale, rotation, and spherical harmonics coefficients. Given a succinct instruction $Q_{inst}$, this task aims to predict an ordered sequence of $T$ binary affordance masks $\mathcal{M} = (M_1, M_2, ..., M_T)$. Each mask $M_t \in \{0, 1\}^N$ identifies the subset of Gaussians that constitute the functional region for the $t$-th atomic instruction, which is implicitly defined by the instruction's action plan, as illustrated in Fig. 1 (left). The objective of this task is to find an optimal mapping $F$ that satisfies

$$\mathcal{M} = F(Q_{inst}, \mathcal{G}). \tag{1}$$

Table 1: Comparison against existing 3D Affordance datasets.

| Benchmark | Vision Type | Scene-level Support | Sequence Support | #Object Cat. | #Afford. Type | #Scenes | #Instances | #Instructions |
|---|---|---|---|---|---|---|---|---|
| 3D-AffordanceNet (CVPR'21) | PointCloud | × | × | 23 | 17 | – | 56k | – |
| LASO (CVPR'24) | PointCloud | × | × | 23 | 17 | – | 8.3k | 1k |
| SeqAfford (CVPR'25) | PointCloud | × | ✓ | 23 | 18 | – | 18k | 40k |
| 3DAffordSplat (ACM MM'25) | Gaussians | × | × | 21 | 18 | – | 23k | 1k |
| **Ours** | **Gaussians** | ✓ | ✓ | **21** | **18** | **1.8k** | **5.6k** | **8k** |

This formulation extends the traditional affordance segmentation paradigm from identifying *what* affordances exist to reasoning about *in what order* they must be actualized to fulfill a user's complex intent.

## 3.2 SeqAffordSplat Dataset Collection

To facilitate research into long-horizon, scene-level affordance reasoning, we introduce SeqAffordSplat, the first large-scale benchmark designed to evaluate sequential affordance grounding directly on 3DGS representations. The construction of the SeqAffordSplat benchmark is a two-stage process meticulously designed to ensure high fidelity and ecological validity across its two core components: the 3D scene geometry, and the sequential, language-grounded instructions.

**Step 1: 3DGS Data Collection.** The foundation of our benchmark lies in the quality and complexity of its 3D environments. To properly evaluate long-horizon reasoning, which often involves interactions among multiple objects, single-object models are insufficient. To generate realistic environments, we manually composed scenes by positioning multiple objects from the 3DAffordSplat (Wei et al., 2025) dataset using geometric transformations. Each scene was constructed based on a specific real-world context (e.g., an office), ensuring that the selection and placement of objects were logical and authentic.

**Step 2: Instruction and Affordance Annotation.** Rather than annotating from scratch, we transferred affordance labels from established benchmarks, primarily 3DAffordSplat (Wei et al., 2025), which provides dense, point-wise affordance labels for thousands of object shapes. The transfer was performed via a semi-automated pipeline: we programmatically match object instances in our scenes with annotated categories in 3DAffordSplat, project the point-wise labels onto our 3DGS representation, and then used a custom 3D annotation tool for manual verification. For a sequential instruction, the ground truth is stored as an ordered list of affordance masks, explicitly encoding the temporal and causal order required for the task. To generate a large and diverse set of long-horizon instructions, we utilized the multimodal large language model (MLLM) GPT-4o (Achiam et al., 2023). We employed a sophisticated prompt engineering strategy that provides the LLM with rich context for each scene, including Role Prompting, Textual Context, Goal Specification, Visual Context and Examples. The generated instruction-sequence pairs underwent a final human-in-the-loop curation process to ensure they are logical, physically possible, and correspond to available affordances.

As shown in Table 1, the final benchmark contains over 1,800 unique 3DGS scenes, annotated with over 14,000 distinct affordance masks across 21 object categories and 18 affordance types. The language component features approximately 8,000 instructions. A key characteristic is its focus on long-horizon tasks: over 47% of instructions require three or more sequential actions, providing a challenging testbed for planning capabilities. Additional details are provided in the Appendix A.2.

## 3.3 Evaluation Configurations

Inspired by the evaluation settings in prior work (Yu et al., 2025), we establish three distinct configurations to comprehensively evaluate our method:

- *Single*: Evaluates the model's ability to predict individual, unordered affordance regions.
- *Sequential (with GT seq)*: Assesses affordance grounding accuracy given a ground-truth action sequence.

Figure 2: An overview of the proposed **SeqSplatNet** architecture. The architecture comprises three main components: a Large Language Model, a 3DGS Encoder with Conditional Geometric Reconstruction Pre-training, and a Conditional Affordance Decoder with VFM Semantic Feature Injection.

- *Sequential*: Tests the full task, where the model must infer and execute the entire action sequence from a single high-level instruction.

This multi-faceted approach allows us to isolate and robustly evaluate the model's core capabilities of sequential reasoning and affordance grounding.

## 4 SEQSPLATNET

### 4.1 ARCHITECTURE

Our SeqSplatNet features an end-to-end architecture that directly maps language instructions to sequential 3D affordance masks. Through an autoregressive process, the model generates interleaved language tokens and special <SEG> tokens, where each <SEG> emission dynamically triggers the affordance decoder to produce a 3D affordance mask. This design inherently unifies task planning and localization by embedding action sequencing within the generative process, eliminating explicit hierarchical decomposition.

As illustrated in Fig. 2, our SeqSplatNet comprises three core components: a 3DGS Encoder, a Large Language Model (LLM) and a Conditional Affordance Decoder, collectively constituting our base model. We augment this framework with two key enhancements: Conditional Geometric Reconstruction Pre-training for improved 3DGS Encoder initialization, and VFM Semantic Feature Injection to enrich geometric representations with semantic knowledge extracted from 2D VFMs.

**3DGS Encoder.** We adopt a PointNet-based encoder (Qi et al., 2017) to extract geometric information from a 3DGS scene $\mathcal{G}$. Consistent with 3DAffordSplat (Wei et al., 2025), our encoder processes the geometric attributes (`position`, `rotation` and `scale`) of Gaussian primitives in $\mathcal{G}$, generating point-wise geometric features $F_{\text{geo}} \in \mathbb{R}^{N \times d}$.

**LLM.** Our LLM serves as the central reasoning engine, processing an input instruction $Q_{\text{instr}}$ to autoregressively generate a primitive instruction sequence. Inspired by recent advancements in Multimodal LLM (Li et al., 2024b; Wei et al., 2025), we augment the token vocabulary with a special token <SEG>. Within the interleaved sequence of language tokens and <SEG> tokens, each <SEG> simultaneously activates affordance mask decoding for its associated primitive instruction and provides a dynamic instruction vector $h_{\text{seg}} \in \mathbb{R}^d$ derived from its hidden state. Benefiting from the masked attention mechanism in LLM, this vector effectively encodes the contextual dependencies from $Q_{\text{instr}}$ and its preceding primitives.

**Conditional Affordance Decoder.** This decoder generates the affordance mask conditioned on each obtained dynamic instruction vector $h_{\text{seg}}$. Built upon recent query-based segmentation

paradigm (Cheng et al., 2021), it employs each LLM-derived dynamic instruction vector $h_{\text{seg}}$ as a latent query to decode its corresponding 3D affordance mask $M_t$ from the encoded geometric feature and injected semantic information (as detailed subsequently).

This tight integration of reasoning and perception within a unified autoregressive framework enables end-to-end sequential reasoning in complex 3DGS scenes with our SeqSplatNet.

**End-to-end Training of SeqSplatNet.** Our SeqSplatNet aims to generate fine-grained affordance masks associated with reasoned primitive instruction sequences for succinct input instructions. To this end, its overall loss summarizes both the language-modeling misalignment $\mathcal{L}_{\text{lang}}$ and the affordance segmentation error $\mathcal{L}_{\text{mask}}$:

$$\mathcal{L}_{\text{total}} = \mathcal{L}_{\text{lang}} + \frac{\lambda_{\text{mask}}}{T} \sum_{t=1}^{T} \mathcal{L}_{\text{mask}}, \tag{2}$$

where $\lambda_{\text{mask}}$ is a balancing hyperparameter. In this work, we adopt the standard autoregressive cross-entropy loss over the predicted token sequence for $\mathcal{L}_{\text{lang}}$. The segmentation loss $\mathcal{L}_{\text{mask}}$, activated at each <SEG> token, is a combination of Binary Cross-Entropy (BCE) and Dice losses to ensure both pixel-wise accuracy and structural similarity:

$$\mathcal{L}_{\text{mask}} = \mathcal{L}_{\text{BCE}}(\hat{M}_t, M_t^{\text{gt}}) + \mathcal{L}_{\text{Dice}}(\hat{M}_t, M_t^{\text{gt}}), \tag{3}$$

where $\hat{M}_t$ and $M_t^{\text{gt}}$ are the predicted and ground-truth masks for step $t$, respectively.

## 4.2 Conditional Geometric Reconstruction Pre-training

Training an effective 3DGS encoder from scratch is challenging for complex scenes due to significant annotation requirements. To address this, we introduce a pre-training strategy, depicted in Fig. 2(b), designed to instill a robust geometric prior in our 3DGS encoder. The core idea is to train the encoder on a pretext task that forces it to understand geometry functionally: it must learn to reconstruct an affordance region guided solely by an abstract semantic concept.

Specifically, given a 3DGS scene $\mathcal{G}$ and an affordance mask $M^{\text{gt}} \in \{0, 1\}^N$, our pre-training framework consists of three components: the main 3DGS Encoder $\Phi_{\text{enc}}$, a lightweight Mask Encoder $\Phi_{\text{mask}}$, and a Decoder $\Phi_{\text{dec}}$. The 3DGS encoder $\Phi_{\text{enc}}$ extracts per-point features $F_{\text{geo}}$ from $\mathcal{G}$, while the mask encoder $\Phi_{\text{mask}}$ embeds $M^{\text{gt}}$ into a semantic condition vector $e_{\text{mask}} \in \mathbb{R}^d$. Mask Reconstruction is then conditioned on $e_{\text{mask}}$ via a cross-attention mechanism,

$$F_{\text{fused}} = \text{Attention}(Q = e_{\text{mask}}, K = F_{\text{geo}}, V = F_{\text{geo}}), \tag{4}$$

$$\hat{M} = \Phi_{\text{dec}}(F_{\text{fused}}). \tag{5}$$

where $e_{\text{mask}}$ queries the geometric features $F_{\text{geo}}$ to yield aggregated features $F_{\text{fused}}$, and the decoder $\Phi_{\text{dec}}$ subsequently transforms them into the predicted mask $\hat{M}$.

In this work, we optimize this pre-training model by minimizing a composite reconstruction loss $l_{rec}$, which combines binary cross-entropy and Dice loss between the predicted $\hat{M}$ and ground-truth $M^{\text{gt}}$ masks. Our pre-training strategy compels the network to learn a powerful mapping from geometric structures to affordance concepts, providing improved initialization for our 3DGS encoder.

## 4.3 VFM Semantic Feature Injection

Interpreting nuanced language instructions to identify affordances requires a deep semantic understanding that pure geometric representations cannot provide for complex scenes. To bridge this gap, we leverage the high-fidelity rendering capability of 3DGS to inject potent semantic knowledge from pre-trained 2D Vision Foundation Models (VFM).

For a scene represented by a set of $n$ 3D Gaussians, we first generate $m$ multi-view 2D feature maps $\{F^{(v)}\}_{v=1}^{m}$. Each feature map is obtained by processing a rendered RGB image $I^{(v)}$ with a frozen, pre-trained VFM, $\Psi_{\text{VFM}}$ (e.g., DINOv2 (Oquab et al., 2023), CLIP (Radford et al., 2021)):

$$F^{(v)} = \Psi_{\text{VFM}}(I^{(v)}) \in \mathbb{R}^{H \times W \times d_{\text{sem}}}. \tag{6}$$

To lift these 2D features into the 3D space, we employ a learning-free aggregation process akin to an inverse rendering operation following the learning-free lifting paradigm (Marrie et al., 2024). This approach correctly handles the contribution of multiple Gaussians to each rendered pixel via alpha-blending. The semantic feature vector $f_i^{\text{sem}}$ for each Gaussian $i$ is computed as a weighted average of all the 2D pixel features it influences across all views. The weight for each pixel feature $F_p^{(v)}$ (from view $v$ at pixel $p$) is its rendering weight $w_i(v, p)$, which represents the influence of Gaussian $i$ on that pixel. The lifted feature is defined as:

$$f_i^{\text{sem}} = \frac{\sum_{(v,p) \in \mathcal{S}_i} w_i(v,p) F_p^{(v)}}{\sum_{(v,p) \in \mathcal{S}_i} w_i(v,p)}, \tag{7}$$

where $\mathcal{S}_i$ is the set of all view-pixel pairs $(v, p)$ that Gaussian $i$ contributes to. This aggregation method inverts the rendering process to produce a semantically-rich feature bank $F_{\text{sem}} \in \mathbb{R}^{n \times d_{\text{sem}}}$ attached to the Gaussians.

Subsequently, the lifted semantic features $F_{\text{sem}}$ are injected into each layer of our Conditional Affordance Decoder via additive fusion. This multi-layer strategy enhances semantic consistency in segmentation by informing the decoding process at all levels of granularity, from coarse to fine-grained.

## 5 EXPERIMENTS

### 5.1 EXPERIMENTAL SETTINGS

**Baseline Models.** Since our method is the first sequence reasoning approach based on 3DGS, for a fair comparison, we selected the following baselines: AffordSplatNet (Wei et al., 2025), a single-step reasoning method based on 3DGS; PointRefer (Li et al., 2024b) and IAGNet (Yang et al., 2023), single-step methods based on point clouds; and SeqAfford (Yu et al., 2025), a sequence reasoning method based on point clouds.

**Evaluation Metrics.** For single-step prediction, we adopt standard metrics **mIoU**, **AUC**, **SIM**, **MAE** to ensure a fair comparison with prior works like LASO (Li et al., 2024b). For sequential tasks, we introduce **sIoU**, **sAUC**, **sSIM**, and **sMAE**, which extend standard metrics to evaluate the entire generated sequence, thus assessing both spatial and temporal accuracy. Detailed definitions are in the Appendix A.3.1.

**Implementation Details.**

For our primary results, we use Qwen-3-0.6B (Yang et al., 2025) as our LLM, fine-tuned with LoRA (Hu et al., 2022) (rank=8, alpha=16) targeting the q_proj, k_proj, v_proj, and lm_head layers. The model is pre-trained for 10 epochs, followed by 50 epochs of end-to-end training where non-LLM parameters use a decaying learning rate. The Adam optimizer (Kingma & Ba, 2014) with weight decay is used, and we set $\lambda_{\text{mask}} = 1$. Experiments were conducted on 8 GeForce RTX 3090 GPUs. Additional details are provided in the Appendices A.1.

### 5.2 RESULTS ON SEQAFFORDSPLAT DATASET

We performed the Sequential 3D Gaussian Affordance Reasoning task on the SeqAffordSplat Dataset. As outlined in the Dataset section, this task is categorized into three distinct settings based on the nature of the instructions: *Single*, *Sequential (with GT sequence)*, and end-to-end *Sequential*. The main results are presented in Table 2.

**Single.** In the single-step setting, models predict an affordance mask for a single, explicit instruction. Our method achieves the state-of-the-art performance with an *mIoU* of 37.0% and an *AUC* of 94.0%. This represents a significant improvement of 5.7% in *mIoU* over the strongest point-cloud-based baseline, PointRefer (31.3%), and 6.5% improvement over the 3DGS-based baseline, AffordSplatNet (30.5%). Notably, the performance of AffordSplatNet is slightly lower than that of PointRefer. A possible reason is that AffordSplatNet treats the 3D Gaussians simply as a point cloud enriched with features like opacity, scale, and rotation, without fully leveraging the high-fidelity rendering capabilities inherent to 3DGS. This demonstrates the superior capability of our architecture in understanding and grounding instructions even in non-sequential scenarios.

Table 2: Results on SeqAffordSplat dataset

| Main results | Method | Source | mIoU/sIoU↑ | AUC/sAUC↑ | SIM/sSIM↑ | MAE/sMAE↓ |
|---|---|---|---|---|---|---|
| **Single** | AffordSplatNet | ACM MM'25 | 30.5 | 92.7 | 0.395 | 0.065 |
| | PointRefer | CVPR'24 | 31.3 | 92.1 | 0.411 | 0.055 |
| | IAGNet | ICCV'23 | 17.6 | 85.2 | 0.328 | 0.056 |
| | **OURS** | - | **37.0** | **94.0** | **0.470** | **0.049** |
| **Sequential (with GT seq)** | AffordSplatNet | ACM MM'25 | 26.1 | 91.2 | 0.343 | 0.072 |
| | PointRefer | CVPR'24 | 30.3 | 91.2 | 0.418 | 0.055 |
| | IAGNet | ICCV'23 | 13.9 | 88.0 | 0.325 | 0.062 |
| | **OURS** | - | **36.0** | **95.6** | **0.457** | **0.036** |
| **Sequential** | SeqAfford | CVPR'25 | 12.1 | 73.0 | 0.122 | 0.230 |
| | **OURS** | - | **26.2** | **80.6** | **0.312** | **0.132** |

Table 3: Results on 3DAffordSplat dataset

| Method | mIoU↑ | AUC↑ | SIM↑ | MAE↓ |
|---|---|---|---|---|
| AffordSplatNet | 30.3 | 83.9 | 0.440 | 0.210 |
| IAGNet | 14.6 | 56.7 | 0.350 | 0.410 |
| PointRefer | 18.4 | 78.5 | 0.430 | 0.200 |
| **OURS** | **40.2** | **89.3** | **0.530** | **0.169** |

**Sequential (with GT seq).** This setting evaluates the model's ability to ground affordances given a ground-truth sequence of sub-instructions. This isolates the performance of the perception module from the language reasoning component. Our method continues to outperform all baselines, achieving an *sIoU* of 36.0%. This is 5.7% higher than the next-best baseline, PointRefer, indicating that our conditional decoder excels at accurately interpreting specific sub-tasks and generating precise affordance masks.

**Sequential.** This is the full end-to-end task, where the model must reason about a complex instruction and generate the entire sequence of affordance masks. Since other baselines do not support end-to-end sequential reasoning, we compare our method against SeqAfford (Yu et al., 2025), the only available baseline for this task, which operates on point clouds. In this challenging setting, our method demonstrates a remarkable improvement. We achieve an *sIoU* of 26.2%, which is more than double the performance of SeqAfford (12.1%). This substantial gain of 14.1% points in *sIoU* underscores the effectiveness of our integrated reasoning and perception framework, which leverages the LLM to decompose tasks and the decoder to ground them in the 3DGS representation.

## 5.3 RESULTS ON 3DAFFORDSPLAT DATASET

To validate the generalization capability of our approach, we also evaluated it on the existing 3DAffordSplat dataset (Wei et al., 2025), which focuses on single-step affordance reasoning on 3D Gaussian data. As shown in Table 3, our method achieves an *mIoU* of 40.2%, significantly outperforming all prior methods. This result is 9.9% higher than the original 3DAffordSplat benchmark, confirming that the architectural designs and training strategies proposed in our work are robust and effective beyond our new sequential task, setting a new state-of-the-art on this established benchmark as well.

Table 4: Ablation study of main components

| Component | | sIoU↑ | sAUC↑ | sSIM↑ | sMAE↓ |
|---|---|---|---|---|---|
| Pretrain | Feature | | | | |
| × | × | 20.3 | 76.3 | 0.229 | 0.169 |
| ✓ | × | 24.1 | 78.5 | 0.302 | 0.141 |
| ✓ | CLIP | 24.2 | 79.1 | 0.290 | 0.141 |
| ✓ | DINOv2 | **26.2** | **80.6** | **0.312** | **0.132** |

Table 5: Ablation study of LLM backbones

| LLM | sIoU↑ | sAUC↑ | sSIM↑ | sMAE↓ |
|---|---|---|---|---|
| GPT2-small(0.1B) | 12.1 | 43.9 | 0.156 | 0.488 |
| **Qwen3-0.6B** | 26.2 | **80.6** | **0.312** | **0.132** |
| Qwen3-1.7B | **26.4** | 79.5 | 0.291 | 0.147 |
| Qwen3-8B | 24.2 | 78.6 | 0.285 | 0.148 |

## 5.4 ABLATION STUDY

We conducted ablation studies to analyze the contribution of each key component in our framework on *Sequential* task.

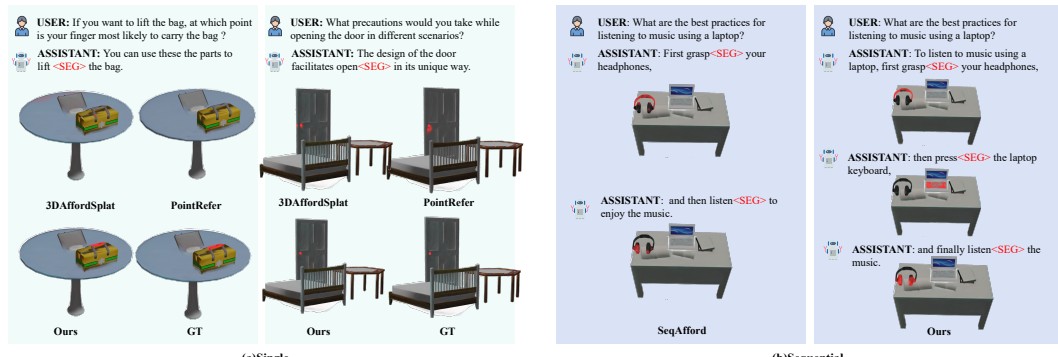

Figure 3: Visual Results of our proposed methods. The predicted regions are highlighted in red.

**Effects of Different Components.** As shown in Table 4, our SeqSplatNet base model, constructed with the three key components (a 3DGS Encoder, an LLM and a Conditional Affordance Decoder), outperforms the pioneering SeqAfford by 8.2% (20.3% *v.s.* 12.1%) in *sIoU*. Integration of our Conditional Geometric Reconstruction Pre-training strategy elevates the *sIoU* performance to 24.1%, then activating our VFM Semantic Feature Injection with DINOv2 further increases it to 26.2%. This validates the effectiveness of our base model, pre-training strategy and semantic feature injection.

**Effects of Different LLM Encoders.** We investigated the impact of the different LLMs on reasoning performance, as summarized in Table 5. The GPT2-small (0.1B) (Radford et al., 2019) model serves as a baseline, achieving an *sIoU* of only 12.1%, which underscores the necessity of a more capable language model. Among the Qwen3 series, our primary model, Qwen3-0.6B, performs very competitively with an *sIoU* of 26.2%, while also achieving the best *sAUC* (80.6%) and *sSIM* (0.312) scores. Interestingly, the largest model tested, Qwen3-8B, shows a performance degradation to 24.2% in *sIoU*, suggesting that simply increasing parameter count does not guarantee better performance on sequential affordance reasoning. This result likely stems from the 8B model overfitting on our specialized dataset, where its vast capacity causes it to memorize training artifacts instead of general logic. We select Qwen3-0.6B for our main experiments as it achieves state-of-the-art performance with excellent model efficiency.

## 5.5 QUALITATIVE RESULTS

Fig. 3 presents a qualitative comparison of our method against baselines. In single-step scenarios (a), our model demonstrates superior precision. For example, it correctly identifies the specific liftable part of a bag based on a nuanced instruction, while competing methods segment the incorrect region. More critically, for the sequential task (b), our model successfully decomposes a high-level command ("listen to music using a laptop") into a logical, multi-step sequence of affordances. In contrast, the baseline method fails to generate a coherent plan, visually validating our framework's advanced reasoning and grounding capabilities for complex, long-horizon tasks. More qualitative results, including experiments on *real-world* scenes, are provided in the Appendices A.4 and A.5.

## 6 CONCLUSION

We advance 3D affordance reasoning on 3DGS from single-step interactions to complex, sequential scene-level tasks. To this end, we introduce the SeqAffordSplat benchmark and SeqSplatNet, the first framework unifying LLM planning with high-fidelity 3DGS representations. Our model, using novel geometric pre-training and semantic injection, sets a new state-of-the-art with a 14.1% performance gain on our benchmark. This work provides a critical foundation for developing more capable embodied agents that can understand and execute the long-horizon, multi-step instructions characteristic of real-world environments.

## 7 REPRODUCIBILITY STATEMENT

To ensure the reproducibility of our work, all implementation details required to replicate our findings are thoroughly documented within this paper. This includes comprehensive descriptions of our model architecture, hyperparameters and training procedures (Appendix A.1), the methodology for our dataset's creation (Appendix A.2), and the complete experimental settings for all evaluations (Appendix A.3). We commit to making our entire source code and the SeqAffordSplat dataset publicly available upon publication.

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

# A APPENDIX

CONTENTS

## A.1 METHOD DETAILS

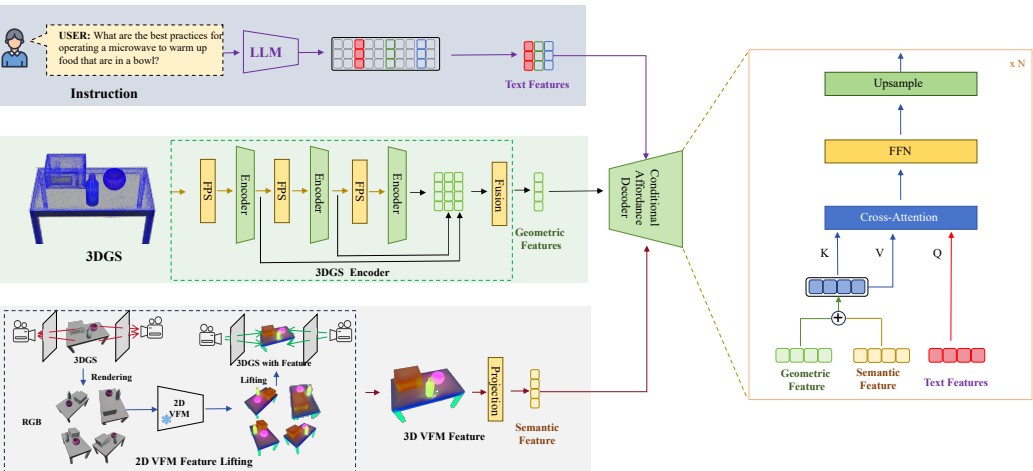

Figure 4: Detailed pipeline of our proposed SeqSplatNet framework.

### A.1.1 IMPLEMENTATION DETAILS OF KEY COMPONENTS

This section provides specific implementation details for the core components of SeqSplatNet, supplementing the architectural overview in the main paper. A detailed pipeline is visualized in Fig. 4.

**3DGS Encoder.** Our encoder employs a hierarchical structure with three set abstraction levels. At each level, we use Farthest Point Sampling (FPS) to downsample the Gaussian primitives into subsets of 2048, 512, and 128 points, respectively. For feature aggregation across different scales,

we adopt the same approach as AffordSplatNet (Wei et al., 2025). Specifically, features from all levels are upsampled to the original resolution of $N$ points via inverse distance weighted (IDW) interpolation and are then concatenated.

**LLM.** For the various LLMs tested, we format inputs using their respective official chat templates to ensure optimal performance. The hidden state of the LLM corresponding to an emitted <SEG> token is extracted to form the dynamic query vector $h_{seg}$ for the affordance decoder.

**Conditional Affordance Decoder.** The decoder consists of a stack of $L = 3$ transformer decoder layers. The query vector $h_{seg}$ attends to the additively fused geometric and semantic features. The output is progressively refined through Feed-Forward Networks (FFNs) and upsampling blocks. Finally, a parameter network generates dynamic convolution kernels, which are convolved with the full-resolution features to produce the mask logits, followed by a sigmoid function to yield the binary mask.

**Conditional Geometric Pre-training.** This stage uses a specialized Mask Encoder ($\Phi_{mask}$) and a lightweight decoder ($\Phi_{dec}$).

- The **Mask Encoder**, detailed in Fig. 5, downsamples points via FPS, forms local groups, and uses a 1D CNN with positional embeddings and an attention mechanism to produce a single condition vector $e_{mask}$.
- The **lightweight decoder** is a simplified version of our main decoder containing only 2 cross-attention layers, and it is optimized to reconstruct the original mask conditioned on $e_{mask}$.

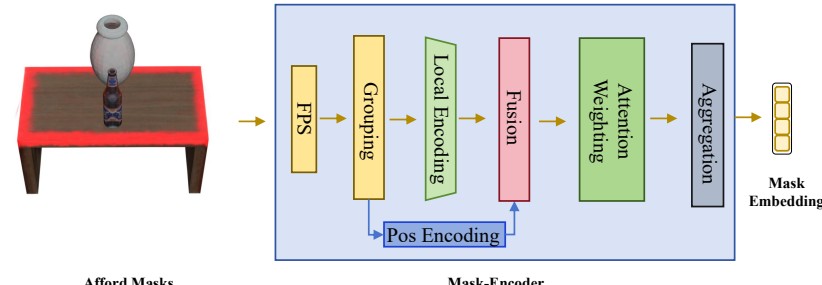

Figure 5: The architecture of the Mask Encoder used in our Conditional Geometric Reconstruction Pre-training stage.

**VFM Semantic Feature Injection.** To generate semantic features, we render the 3DGS scene from 100 camera positions uniformly sampled on the upper hemisphere. Each image is rendered at a 640x480 resolution and then processed by a frozen DINOv2 model. The resulting 2D feature maps are lifted to 3D Gaussians using the learning-free weighted averaging technique described in the main paper.

### A.1.2 TRAINING AND INFERENCE PROCEDURE

Our framework's training and inference procedures, while sharing the same architecture, operate in distinct modes.

**Training.** We employ a teacher-forcing strategy. The model is fed the ground-truth sequence, including both language instructions and answer text with special <SEG> tokens. The mask decoding process is triggered exclusively by the ground-truth <SEG> tokens, using their corresponding hidden states as queries. This design simplifies the loss calculation, while the language modeling loss ($\mathcal{L}_{lang}$) implicitly penalizes deviations in sequence order and length.

**Inference.** The model operates in an autoregressive mode, generating the output sequence one token at a time. The Conditional Affordance Decoder is activated whenever the model emits a <SEG> token. In scenarios where the model fails to generate any <SEG> tokens, we adopt a fallback strategy: a single affordance mask is decoded using a query vector derived from the average of the hidden states of all generated text tokens.

## A.2 DATASET DETAILS

### A.2.1 DATA COLLECTION

Our `SeqAffordSplat` dataset is built upon the object-level assets from the existing 3DAfford-Splat dataset (Wei et al., 2025). The collection process consists of two primary stages: 3DGS scene composition and the annotation of sequential instructions and affordances. To ensure diversity and real-world relevance, we first defined six thematic categories for our scenes: office/workspace, kitchen/dining, electronic device usage, living room, storage/organization, and bedroom.

**3DGS Scene Composition.** For each scene category, we first determined plausible object combinations based on real-world contexts (e.g., a desk, chair, and laptop for an office scene). We then selected the corresponding object-level 3DGS assets from the 3DAffordSplat dataset. Following principles of realistic object placement, distribution, and scale, we manually composed the final scenes. As illustrated in Fig. 6, this involved applying a series of geometric transformations—specifically translation, rotation, and scaling—to arrange the individual objects into a coherent and often cluttered environment. This bottom-up approach allowed us to generate a large corpus of over 1,800 complex, multi-object 3DGS scenes that are essential for evaluating long-horizon sequential reasoning tasks.

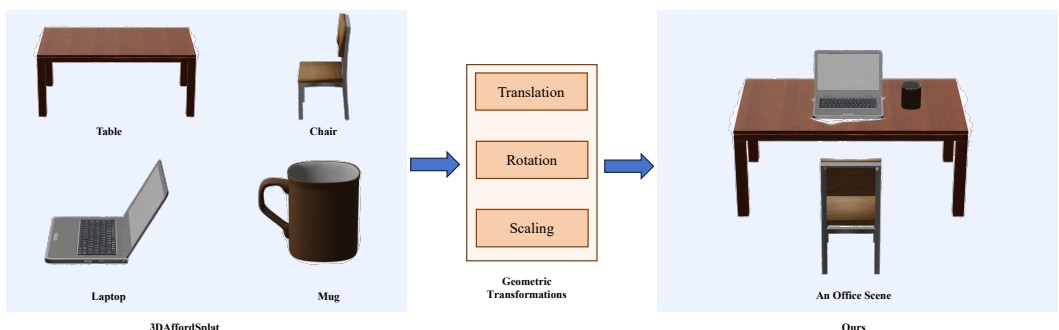

Figure 6: The pipeline for composing complex 3DGS scenes from individual object assets.

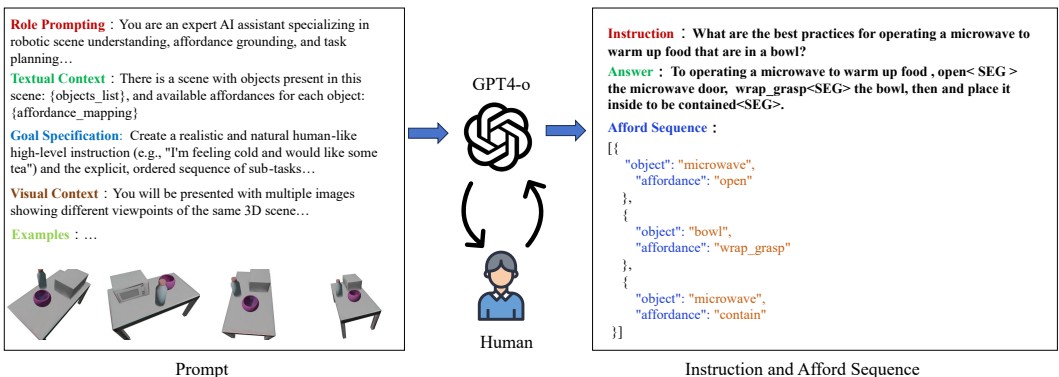

Figure 7: The pipeline for collecting sequential instructions and affordance annotations using GPT-4o and a human-in-the-loop process.

**Instruction and Affordance Annotation.** The creation of our instruction-annotation pairs involves two parallel processes: generating the language instructions and annotating the corresponding 3D affordance masks.

For instruction generation, we utilize the multimodal large language model (MLLM) GPT-4o, following a similar strategy to Yu et al. (2025). Our instruction collection pipeline is visualized in Fig. 7. We employ a carefully designed prompt engineering strategy that provides the MLLM with rich context for each scene, including:

- **Role Prompting**: A high-level instruction defining the MLLM's persona (e.g., "You are an expert AI assistant specializing in robotic scene understanding, affordance grounding, and taskplanning.").

- **Textual Context**: A structured list of all interactable objects and their available affordances.

- **Goal Specification**: A high-level goal (e.g., "I'm feeling cold and would like some tea") that requires the MLLM to generate both a single, naturalistic instruction and the explicit, ordered sequence of sub-tasks.

- **Visual Context**: A series of rendered images from multiple viewpoints of the 3DGS scene.

- **Examples**: Several complete demonstrations of the desired input-output format (i.e., a high-level goal paired with the corresponding natural language instruction and structured sub-task sequence) to guide the model's output.

The generated instruction-sequence pairs then undergo a final human-in-the-loop curation process to ensure they are logical, physically possible, and correspond to available affordances.

For the affordance mask annotation, instead of labeling from scratch, we developed a semi-automated pipeline to transfer existing labels. We programmatically transfer the dense, point-wise affordance labels from the original 3DAffordSplat object assets onto our newly composed scenes. This is followed by a manual verification and refinement step using a custom 3D annotation tool to ensure the accuracy and correctness of the final masks. The ground truth for a sequential instruction is then stored as an ordered list of these verified affordance masks, explicitly encoding the required temporal sequence for the task.

### A.2.2 Data Statistics and Analysis

The dataset is partitioned into training, validation, and test sets at an 8:1:1 ratio. This split, visualized in Fig. 8 (Left), ensures a robust and standardized evaluation pipeline.

**Scene Composition and Object Diversity.** Our dataset's diversity is analyzed through its object distribution across different scenes. As shown in Fig. 8 (Right), categories such as `storage`, `office`, and `kitchen` are the most object-rich, providing complex environments for interaction. The Objects word cloud (Fig. 9) reinforces this by showing that foundational items like `table`, `chair`, and `door` are ubiquitous, reflecting realistic furniture arrangements. Meanwhile, the presence of category-specific items (e.g., `faucet` in kitchens, `bed` in bedrooms) confirms the thematic integrity of the scenes.

**Task Characteristics and Complexity.** The dataset is designed to support long-horizon, action-oriented tasks. This is evident in the alignment between the instruction and affordance word clouds (Fig. 9). Common user goals, indicated by words like `water`, `drinking`, and `fill`, are directly supported by a vocabulary of physical affordances, where affordances like `contain`, `grasp`, `open`, and `pour` are most prominent.

Furthermore, the vast majority of tasks require multi-step reasoning, as detailed in the sequence length distribution (Fig. 10). Sequences of length 2 and 3 are dominant across all categories, with more complex tasks (lengths 4-6) also present, particularly in the `kitchen` scenes. This distribution provides a challenging and varied testbed for evaluating sequential reasoning models.

**Logical and Physical Plausibility.** To validate the dataset's ecological validity, we performed a co-occurrence analysis, visualized in Fig. 11. The heatmap of affordance types (Left) reveals strong correlations between functionally related actions. For instance, high co-occurrence is observed between (`grasp`, `open`) and (`grasp`, `pour`), which reflects natural task execution flows like grasping a handle to open a door or grasping a cup to pour water. The object co-occurrence heatmap (Right) further demonstrates that objects are composed realistically, with high co-occurrence rates between items typically found together, such as (`table`, `chair`) and (`laptop`, `keyboard`). This analysis confirms that the scenes and tasks in our dataset are logically and physically plausible.

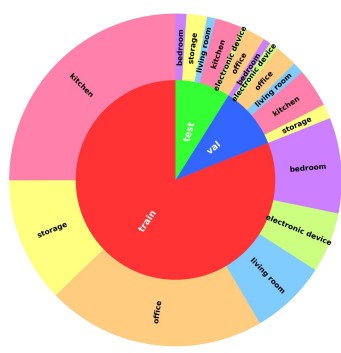 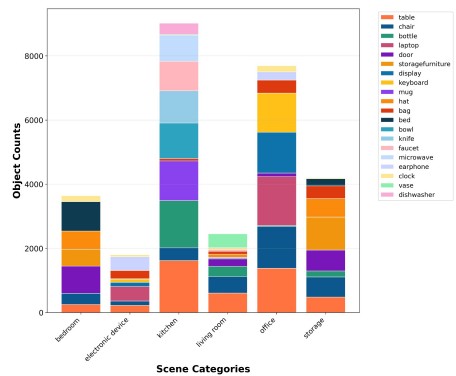

Figure 8: **(Left)** Data Split and **(Right)** Scene-Object Distribution for the SeqAffordSplat dataset.

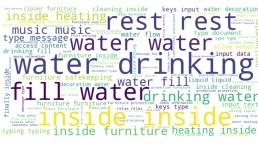

Instructions word cloud

Affordances word cloud

Objects word cloud

Figure 9: Word clouds for objects (left), affordances (center), and instructions (right) in our dataset.

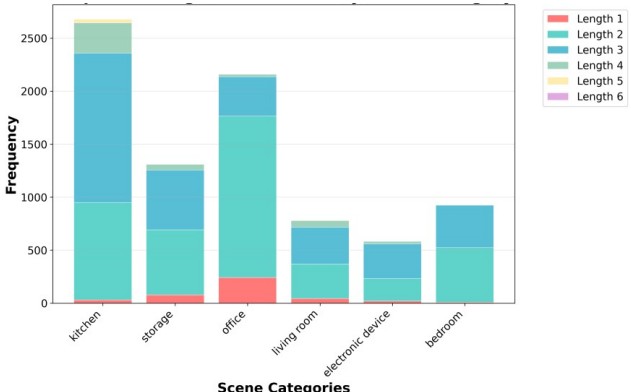

Figure 10: Distribution of instruction sequence lengths across different scene categories.

### A.2.3 ANNOTATED EXAMPLES

Fig.12 showcases several representative examples of our annotated scenes across various categories. This figure illustrates the variety of interactable parts that are labeled in our dataset, which form the basis for the sequential tasks. For instance, an Office scene shows a desk, chair, and laptop, where the colored regions correspond to distinct affordances (e.g., `chair-sit`, `table-support`, `earphone-grasp`). These individual affordances can be chained together to fulfill a complex, high-level instruction.

### A.3 EXPERIMENTAL DETAILS

### A.3.1 EVALUATION METRICS

To rigorously evaluate our model's performance, we define two sets of metrics tailored for single-step prediction and full-sequence prediction, respectively.

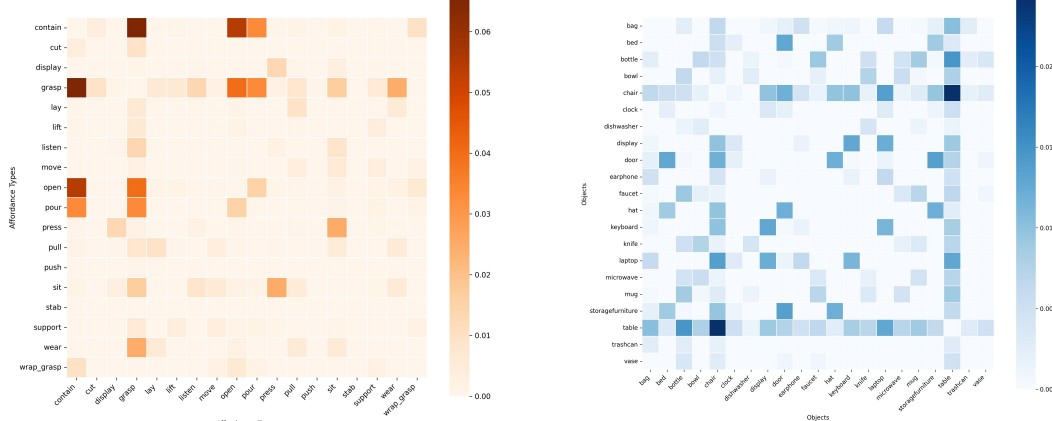

Figure 11: Co-occurrence heatmaps of **(Left)** affordance types and **(Right)** object categories in SeqAffordSplat.

**Single-Step Metrics.** For evaluating predictions at a single, specific timestep, we adhere to the standard metrics established in prior works like LASO (Li et al., 2024b) to ensure a fair and direct comparison. These metrics include:

- **Mean Intersection over Union (mIoU):** Measures the spatial overlap between the predicted mask ($\hat{M}$) and the ground-truth mask ($M^{gt}$), averaged over all test samples.

- **Area Under the Curve (AUC):** Refers to the area under the Precision-Recall curve, evaluating the trade-off between Precision and Recall across all possible classification thresholds.

- **Similarity (SIM):** A metric to assess the structural similarity.

- **Mean Absolute Error (MAE):** Calculates the average absolute pixel-wise difference between the predicted mask and the ground-truth mask.

**Sequential Metrics.** Evaluating an entire sequence requires assessing both the accuracy of each prediction and the temporal coherence of the overall sequence. To this end, we introduce a suite of sequential metrics: **sIoU**, **sAUC**, **sSIM**, and **sMAE**. The core idea is to compute the average metric value over the full temporal span of both sequences after aligning them to a common length, which inherently penalizes discrepancies in sequence duration.

Formally, consider a ground-truth sequence $\mathcal{M}^{gt} = (M_1^{gt}, \ldots, M_{T_{gt}}^{gt})$ and a predicted sequence $\hat{\mathcal{M}} = (\hat{M}_1, \ldots, \hat{M}_{T_{pred}})$. We first align both to a length $T_{max} = \max(T_{gt}, T_{pred})$ by padding the shorter sequence with empty frames ($\emptyset$). Let the padded sequences be $\mathcal{M}'^{gt}$ and $\hat{\mathcal{M}}'$. The sequential IoU (sIoU) is then defined as the average of the IoU across all $T_{max}$ steps:

$$\text{sIoU}(\mathcal{M}^{gt}, \hat{\mathcal{M}}) = \frac{1}{T_{max}} \sum_{t=1}^{T_{max}} \text{IoU}(M_t'^{gt}, \hat{M}_t')$$

The other sequential metrics (sAUC, sSIM, sMAE) are calculated analogously. This approach elegantly punishes incorrect temporal predictions, providing a comprehensive evaluation of both spatial accuracy and temporal extent.

### A.3.2 EXPERIMENTAL SETTINGS

Our model is evaluated across three distinct settings to comprehensively assess its capabilities. The setup for each setting is detailed below.

**Single Setting.** In this setting, models are tasked with predicting a single affordance mask from a single, explicit instruction. To ensure a fair comparison with prior work and test grounding on

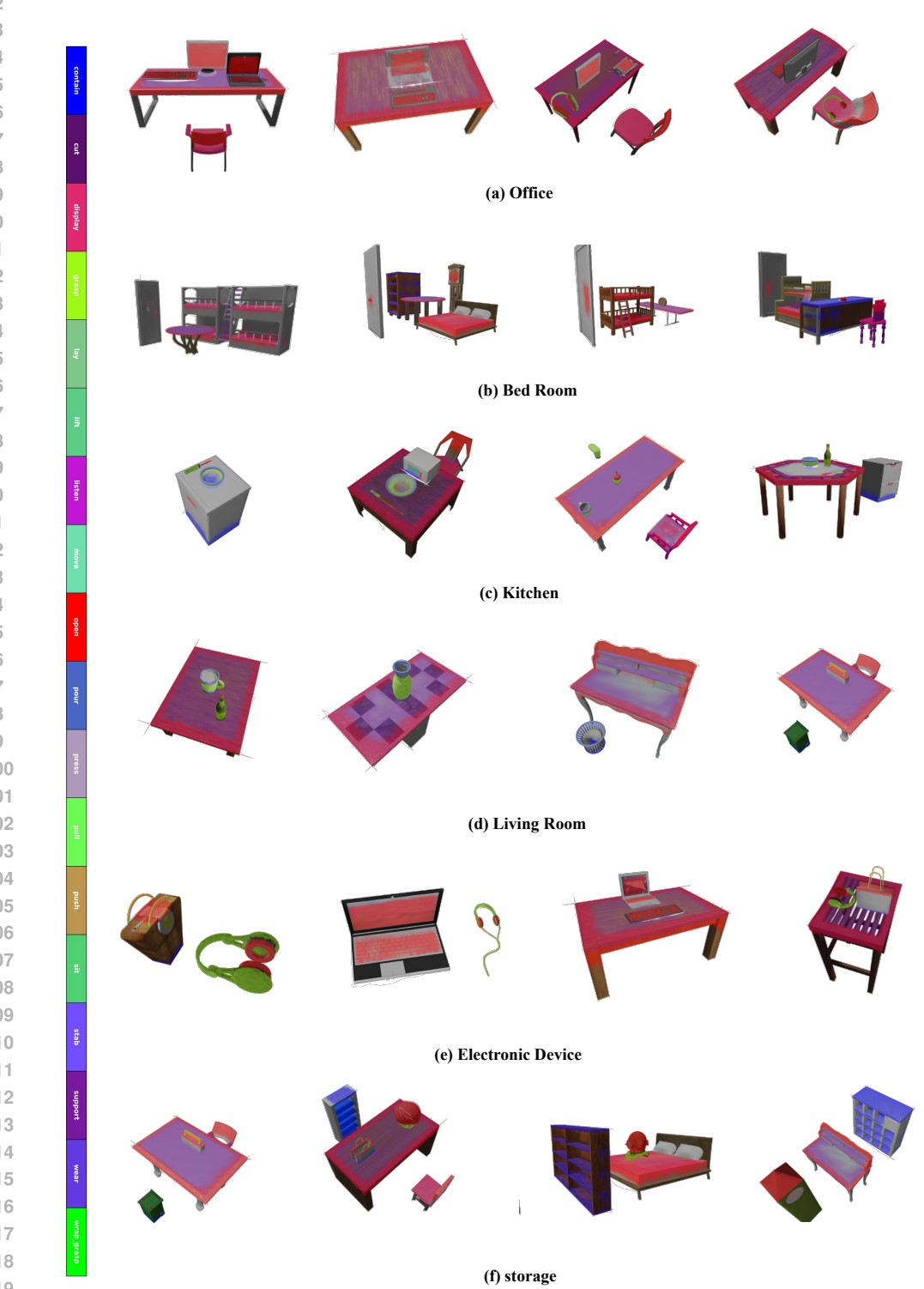

Figure 12: Examples of annotated sequential affordances from our SeqAffordSplat dataset across various scene categories.

concise commands, the language instructions used for evaluation in this setting are sourced directly from the official test set of the LASO dataset (Li et al., 2024b).

**Sequential (with GT seq) Setting.** This setting evaluates a model's ability to ground a sequence of affordances given a ground-truth sequence of sub-tasks, thereby isolating perception from high-level planning. For single-step baselines, which cannot process sequences, we adapt them by treating each sub-instruction in the ground-truth sequence as an independent task. For example, for the high-level task to `"operate a microwave to warm up food that are in a bowl"`, we would query a single-step model three separate times with instructions like `"Best microwave open method?"`, `"How to wrap-grasp the bowl?"`, and `"Ideal contain spot on microwave?"`. The resulting masks are then assembled in order to form the final sequence for evaluation.

**Sequential Setting.** This is the full end-to-end task where the model must interpret a single high-level instruction from our `SeqAffordSplat` dataset and autoregressively generate the entire sequence of affordance masks. This setting evaluates the model's combined planning and perception capabilities. Single-step baselines were not evaluated in this setting.

### A.3.3 BASELINE IMPLEMENTATION DETAILS

To ensure a fair and comprehensive comparison on our `SeqAffordSplat` benchmark, we adapted four state-of-the-art baselines. For all methods, we utilized their official public code and followed the primary hyperparameter settings reported in their original papers, fine-tuning them on our new dataset.

**Data Conversion for Point Cloud Methods.** Since our benchmark is based on 3DGS representation, baselines designed for point clouds required a data conversion step. For `PointRefer` (Li et al., 2024b), `IAGNet` (Yang et al., 2023), and `SeqAfford` (Yu et al., 2025), we converted each 3DGS scene into a point cloud by using the mean XYZ coordinates of the Gaussian primitives.

**Single-Step Baselines.**

- **AffordSplatNet** (Wei et al., 2025): As a 3DGS-native method, it was trained directly on our dataset. Since our composed scenes do not have paired point cloud data, we did not use the point cloud-3DGS alignment pre-training from the original work.
- **PointRefer** (Li et al., 2024b): Trained from scratch on the converted point cloud data without architectural changes.
- **IAGNet** (Yang et al., 2023): Originally image-guided, we adapted it for language instructions by replacing its image backbone with a language model, similar to the strategy in PointRefer (Li et al., 2024b).

**Sequential Baseline.**

- **SeqAfford** (Yu et al., 2025) is the state-of-the-art method for sequential affordance reasoning on point clouds. We fine-tuned it on our converted point cloud data, initializing the model with its pre-trained ShapeLLM (Qi et al., 2024) weights as recommended in the original paper.

### A.4 ADDITIONAL QUALITATIVE RESULTS ON REAL-WORLD SCENES

To further validate the generalization capability and practical applicability of our SeqSplatNet, we tested its performance on several real-world scenes that were captured and reconstructed into 3DGS models. As illustrated in Fig. 13, our model demonstrates a strong ability to understand and ground nuanced instructions on real-world objects. For instance, the model showcases its capacity for physical reasoning by identifying the backrest as the most stable area for moving the chair. These results, with detailed instructions provided in the figure's caption, underscore our model's robustness and its potential for deployment in real-world applications.

### A.5 ADDITIONAL QUALITATIVE RESULTS ON SEQAFFORDSPLAT DATASET

To provide a more intuitive understanding of our model's performance, we present additional qualitative results for both single and sequential settings.

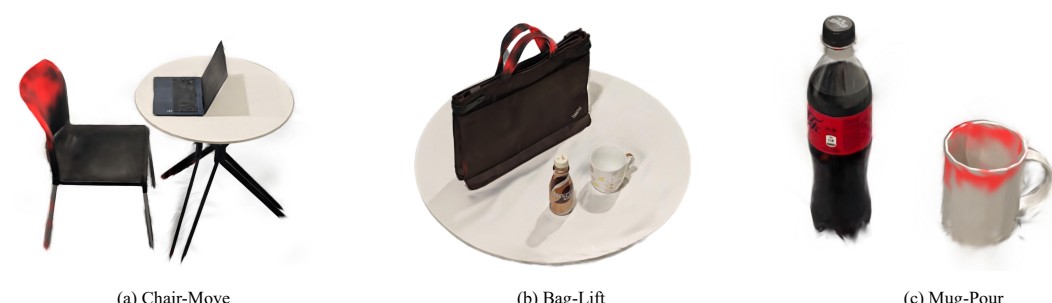

(a) Chair-Move  (b) Bag-Lift  (c) Mug-Pour

Figure 13: Qualitative results of our method on real-world 3DGS scenes. Our model accurately localizes the affordance regions corresponding to detailed instructions, demonstrating strong generalization to unseen, real-world data. The instructions for each sub-figure are: (a) Chair-Move: `"Considering the structure of the chair, what area would be most stable for moving?"`, (b) Bag-Lift: `"What precautions would you take while lifting the bag in different scenarios?"`, and (c) Mug-Pour: `"When emptying water from a mug, from which spot on the mug's edge will the water begin to pour out?"`.

### A.5.1 SINGLE SETTING

Fig. 14 presents qualitative comparisons that highlight our model's superior performance in fine-grained affordance grounding.

For the case `"Which part of the bowl allows for the most efficient wrap_grasping method?"`, our model correctly localizes the entire rim of the bowl, demonstrating a nuanced understanding of the interaction type. In contrast, baselines either identify an incorrect part of the bowl or fail to produce any valid segmentation. Similarly, for the case `"if you want to grab this knife, at which points on the handle will your palm touch?"`, our model successfully identifies the small and precise mask on the handle corresponding to where a palm would make contact, while competing methods fail. These examples underscore our model's robust ability to comprehend complex, descriptive language and translate it into accurate segmentations.

### A.5.2 SEQUENTIAL SETTING

As visualized in Fig.15, our method outperforms the `SeqAfford` baseline in both high-level planning and fine-grained localization. A clear example is (a): the `"Lift the bag and place it on the table for easy access."` case. For this instruction, the ground-truth plan (labeled 'ASSISTANT') consists of three steps. Our model correctly infers this sequence, generating three corresponding affordance masks, which we visualize with explanatory labels such as 'Bag-grasp'. In contrast, the baseline produces a physically implausible sequence by failing to generate a mask for the initial `grasp`. Even in cases where the baseline's plan is coherent, our model provides more accurate mask localization, demonstrating overall superior and more reliable performance for complex sequential tasks.

### A.6 STATEMENT ON THE USE OF LARGE LANGUAGE MODELS

We utilized Large Language Models (LLMs) as an assistive tool to improve the grammar and clarity of this manuscript. This usage is distinct from the LLM that is a core component of our proposed SeqSplatNet framework. The authors take full responsibility for the scientific accuracy and all content of the paper.

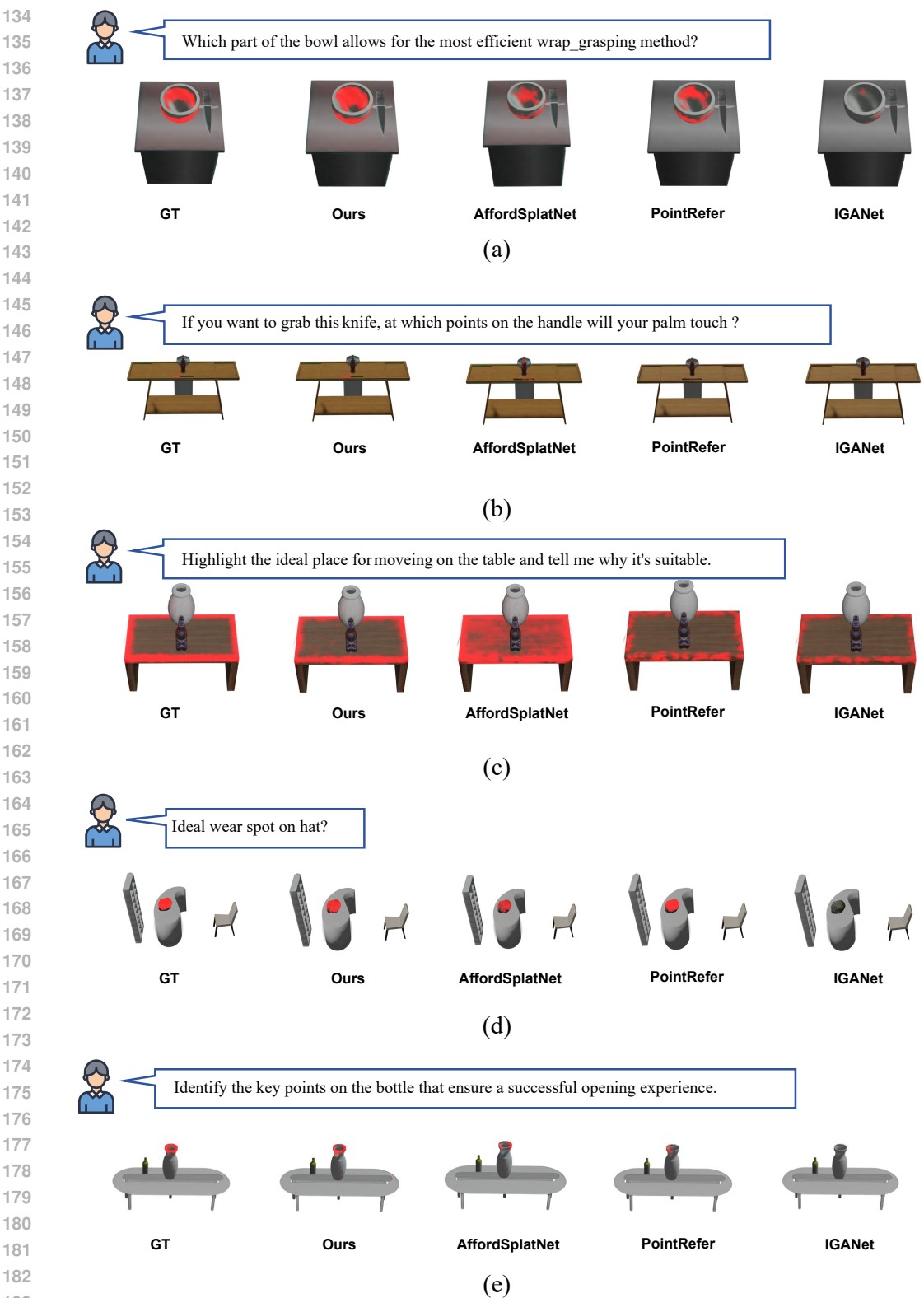

Figure 14: More visual results of the *Single* setting. Our method demonstrates superior precision in grounding nuanced instructions compared to baselines.

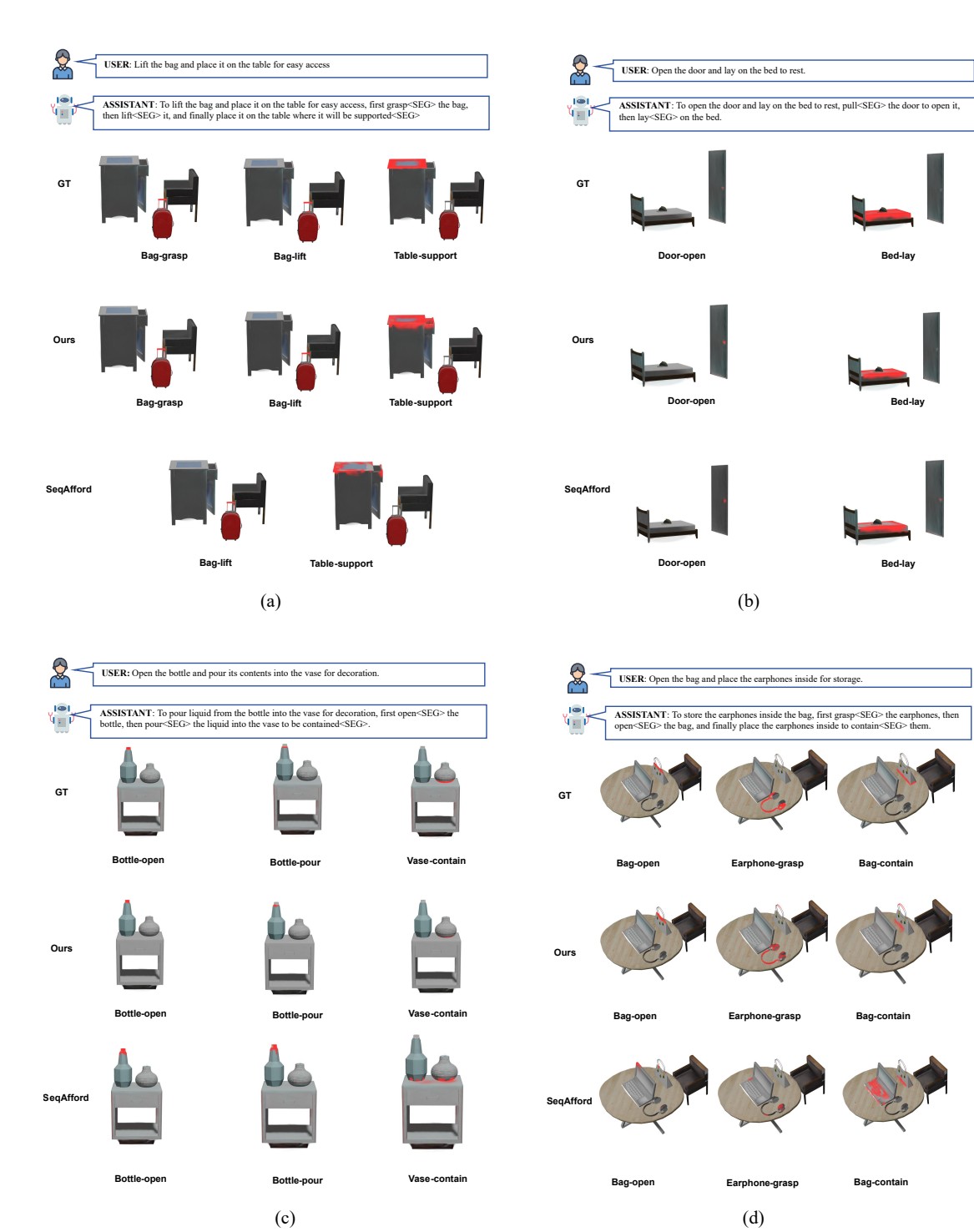

Figure 15: More visual results of the *Sequential* setting. Our method successfully decomposes a high-level command into a logical sequence of affordances, while the baseline fails.

