# OpenReview forum: "SeqAffordSplat: Scene-level Sequential Affordance Reasoning on 3D Gaussian Splatting"
_ICLR.cc/2026/Conference — ICLR 2026 Conference Withdrawn Submission_

### Official Review · Reviewer_npLH · 2025-10-29

**Soundness:** 3
**Presentation:** 3
**Contribution:** 3
**Rating:** 6
**Confidence:** 3

**Summary:**

To address the shortcomings of existing 3D affordance reasoning methods in 3D representation research, which often lack representational capabilities or only support 3DGS representations of single objects and scenes, this paper proposes SeqAffordSplat, a large-scale serialized 3DGS benchmark dataset containing multiple scenes and objects. SeqAffordNet is also introduced, leveraging the reasoning capabilities of LLM end-to-end to directly predict affordance masks in 3DGS scenes using language instructions. On the proposed benchmark, this method achieves a significant improvement over the baseline.

**Strengths:**

1. This paper proposes a large-scale 3D affordance dataset and benchmark, which for the first time supports serialized, scene-level 3D affordance prediction, greatly expanding existing evaluation metrics and providing strong support for related work.
2. At the model level, this paper proposes a solution to the difficulty of convergence in training 3DGS representation encoders from scratch: a pre-training strategy based on conditional geometry reconstruction. By reconstructing affordance masks using the 3DGS encoder, the model gains prior knowledge of how to extract affordance from 3DGS data.
3. In the decoder section, a paradigm is proposed that leverages semantic features extracted from 2D vision models (such as DINOv2), upscaling them to 3D space through multiple views and fusing them with geometric features, further enhancing the model's ability to extract affordance.

**Weaknesses:**

1. The authors point out that part of the scene construction in the dataset needs to be done manually. This process should be automated to increase the dataset's scalability. Also, the paper suggests that the automatic generation of instructions using engineering + GPT-4o followed by manual verification may introduce human bias, leading to data distribution shifts.
2. The paper provides successful visualization examples. Please supplement with typical failure examples and analyze the root causes (LLM errors/occlusion/fine-grained geometric indistinguishability/multi-object confusion, etc.), and provide quantitative statistics (proportion and category of failure examples).

**Questions:**

Did the authors verify the application of the relevant methods in embodied scenarios? For example, can they be transferred to embodied simulator benchmarks (such as LIBERO, SimplerEnv, etc.) and achieve performance improvements?

---

### Official Review · Reviewer_ZK9J · 2025-10-31

**Soundness:** 2
**Presentation:** 3
**Contribution:** 2
**Rating:** 2
**Confidence:** 5

**Summary:**

This paper introduces SeqAffordSplat, a framework for scene-level sequential affordance reasoning based on 3D Gaussian Splatting (3DGS) representations. Unlike prior affordance research that focuses on single objects or isolated actions, this work aims to reason about multi-object, multi-step interactions described in natural language instructions.

To support this, the authors construct a new dataset. SeqAffordSplat. They also propose SeqSplatNet, a model combining:
1/ a language reasoning module (LLM) that predicts ordered sub-actions,
2/ a 3D Gaussian encoder that extracts geometry-aware features,
3/ and an affordance decoder that predicts 3D affordance masks step-by-step.

The method outperforms baselines such as AffordSplatNet, PointRefer, and SeqAfford on both the new dataset and existing 3DAffordSplat benchmark.

**Strengths:**

1/ The model’s design combining a compact LLM (Qwen3-0.6B) with a 3D Gaussian feature encoder is technically sound and aligns with recent multimodal trends. The sequential affordance decoding formulation is clear and well motivated.
2/ The paper is easy to follow, with clear figures and an intuitive overall structure that guides the reader through dataset, model, and results.

**Weaknesses:**

1/ The core framework mostly combines existing components — a pre-trained 3D Gaussian encoder, a small-scale LLM for instruction parsing, and a mask decoder — without introducing fundamentally new algorithms. The idea of sequential reasoning itself is an incremental extension of prior affordance prediction works.
2/ While the dataset is larger and includes sequential labels, it is mainly a compositional expansion of existing resources rather than a new type of data or annotation paradigm. The use of GPT-4o for generating instructions further limits the originality.
3/ The paper claims scene-level reasoning, but experiments primarily measure mIoU/sIoU on localized affordance masks. There is no strong evidence of improved temporal or causal reasoning, only better segmentation accuracy.
4/ The architecture depends heavily on pre-trained foundation models (VFM, LLM, 3DGS encoder). The paper provides limited insights into how these modules interact or why Gaussian Splatting provides unique advantages over standard 3D point features.

**Questions:**

Please see my weakness section.

---

### Official Review · Reviewer_v1bK · 2025-11-01

**Soundness:** 2
**Presentation:** 3
**Contribution:** 2
**Rating:** 2
**Confidence:** 4

**Summary:**

This paper presents a new dataset for sequential 3D affordance reasoning with Gaussian splatting. The model needs to identify a sequence of subtasks and their corresponding affordance masks. To perform the task, the paper proposes to combine LLMs with 3D Gaussian splatting encoder and an affordance decoder that fuses with the lifted semantic features extracted from VFMs. The experiment results show that the proposed method outperforms previous affordance prediction methods.

**Strengths:**

- This is the first work that combines seal task reasoning and affordance prediction. The proposed dataset can be useful for future research.
- The experiments carefully ablate the different components and backbone to demonstrate the contribution of each component choice.

**Weaknesses:**

- It is hard to understand how the inference of the subtask sequence can influence the affordance prediction. In the current presentation, it can be solved by using LLMs to decompose the task into subtasks and then perform single-subtask affordance prediction with Gaussian splatting. Why is a unified LLM needed?
- The proposed method has limited novelty. The 3DGS encoder training and the lifting of semantic features are methods used in prior approaches. The use of LLMs is similar to SeqAfford.

**Questions:**

- For a more complex task, the objects can occlude each other. How does the performance affect by occlusion?
- Since the "Sequential with GT seq" is similar to the concatenation of multiple single sutask, why did the performance for all methods drop?

---

### Note · Authors · 2025-11-12

I have read and agree with the venue's withdrawal policy on behalf of myself and my co-authors.